# Real-Time Recognition and Localization Based on Improved YOLOv5s for Robot’s Picking Clustered Fruits of Chilies

**DOI:** 10.3390/s23073408

**Published:** 2023-03-24

**Authors:** Song Zhang, Mingshan Xie

**Affiliations:** College of Big Data and Information Engineering, Guizhou University, Guiyang 550025, China

**Keywords:** chili recognition, lightweight, 3D location

## Abstract

Chili recognition is one of the critical technologies for robots to pick chilies. The robots need locate the fruit. Furthermore, chilies are always planted intensively and their fruits are always clustered. It is a challenge to recognize and locate the chilies that are blocked by branches and leaves, or other chilies. However, little is known about the recognition algorithms considering this situation. Failure to solve this problem will mean that the robot cannot accurately locate and collect chilies, which may even damage the picking robot’s mechanical arm and end effector. Additionally, most of the existing ground target recognition algorithms are relatively complex, and there are many problems, such as numerous parameters and calculations. Many of the existing models have high requirements for hardware and poor portability. It is very difficult to perform these algorithms if the picking robots have limited computing and battery power. In view of these practical issues, we propose a target recognition-location scheme GNPD-YOLOv5s based on improved YOLOv5s in order to automatically identify the occluded and non-occluded chilies. Firstly, the lightweight optimization for Ghost module is introduced into our scheme. Secondly, pruning and distilling the model is designed to further reduce the number of parameters. Finally, the experimental data show that compared with the YOLOv5s model, the floating point operation number of the GNPD-YOLOv5s scheme is reduced by 40.9%, the model size is reduced by 46.6%, and the reasoning speed is accelerated from 29 ms/frame to 14 ms/frame. At the same time, the Mean Accuracy Precision (MAP) is reduced by 1.3%. Our model implements a lightweight network model and target recognition in the dense environment at a small cost. In our locating experiments, the maximum depth locating chili error is 1.84 mm, which meets the needs of a chili picking robot for chili recognition.

## 1. Introduction

Manual picking of chilies is time-consuming and labor-consuming, and the production cost is very high. Therefore, in order to achieve efficient automatic picking of chili, ensure timely picking of chili fruits, and improve the competitiveness in the chili market, it is particularly important to further study the key technologies of chili picking robots [1]. The intelligent sensing and locating technology of chili information is the key technology of the picking robot. Therefore, to improve the robot’s chili picking efficiency, it is necessary to achieve rapid and accurate recognition and locating of chili fruit targets. In the natural growth environment, chili fruits usually have overlapping occlusion, and the recognition of such fruits is an important issue for the practicality of chili picking robots [2]. The recognition method of chili fruit in overlapping occlusion growth state is different from that in non-occlusion morphology and other growth states. Therefore, the picking robot must be able to automatically recognize chili fruit whether in overlapping occlusion or not. With the rapid development of target recognition and localization technology, these technologies have been widely used in various fields in recent years, and remarkable achievements have been made in the agricultural field, which provides a research basis for chili fruit recognition and localization. Target recognition algorithms are mainly divided into traditional recognition algorithms and deep learning algorithms. The recognition effect of traditional target recognition algorithms will be greatly affected by complex and changeable backgrounds. Researchers have made further improvements on this method. Flores et al. [3] used GooLeNet to distinguish soybean and corn seedlings, but this method had a large number of parameters. In order to realize automatic recognition of apples, Gongal et al. [4] processed Apple images by combining HSI color space and RGB color space, and used Otsu threshold segmentation and Hough transform to segment and recognize apples. The average accuracy rate of identifying apples was 78.9, which was not applicable to chilies with their large numbers and complex environment. Azarmdel et al. [5] applied artificial neural network (ANN) and support vector machine (SVM) to classify mulberries according to their maturity, but this is not suitable for chili growing environments with similar background and target. With the continuous development of deep learning convolutional neural network technology, target recognition and recognition show higher advantages than traditional methods. At present, convolutional neural networks are mainly divided into two categories. One is end-to-end methods [6,7,8], mainly YOLO, SSD and Retinanet. The other is based on region suggestions [9], mainly including Mask RCNN, etc. The latter uses search based target recognition and recognition methods, such as methods based on visual attention and reinforcement learning [10]. Zhu et al. [11] proposed a sugarcane node recognition and spatial locating method based on the YOLOv4 method and a binocular camera. First, sugarcane nodes were identified and detected, and then the three-dimensional coordinate information of the target was obtained by using the binocular camera. Fuentes et al. [12] used three categories of detectors, namely, Faster-RCNN, R-FCN and SSD, to identify and detect tomato pests in real time, using local and global class annotation and data expansion methods. Liu et al. [13] used YOLOv4 to detect cucumbers in the actual environment, built a 13 layer network similar to a VGG model, and extracted image information using two anchors of different scales. Xu et al. [14] cited packet convolution on the basis of SqueezeNet network structure and adopted Channel-shuffle to solve the problem of information non circulation after packet convolution. Li et al. [15] used YOLOv4 to identify green peppers, but the identification results lacked the location information of the peppers.

Existing works have studied chili recognition using different techniques, but there are still problems such as the high number of model parameters and slow detection time, and studies on lightweight chili target recognition algorithms have not been reported. In addition, the chili growth environment is complex, and existing works do not consider the natural growth environment where chilies are obscured. Finally, the existing work on chili fruits lacks the problem of spatial localization ability.

In summary, to address the problems of existing detection models, firstly, we propose a lightweight chili detection model GN-YOLOv5s, which replaces the YOLOv5s backbone network with GhostNet and extracts features from the depth-separable convolution instead of the original convolutional layer, significantly reducing the network computation. Secondly, to further streamline the model size, a regularization term on the scaling coefficients is introduced in the BN layer for sparse training, and channels of lower importance are filtered out and pruned;Finally, in order to maintain the high detection accuracy of the lightweight model, the GNPD-YOLOv5s model was obtained by fine-tuning the pruned model with the aid of a teacher network (the original YOLOv5s model) using a knowledge distillation method. The validity of the GNPD-YOLOv5s model was verified by experiments. Our contributions are as follows:

(1) We propose a GN-YOLOv5s algorithm for real-time chili detection. Based on the original YOLOv5s algorithm, this algorithm introduces GhostNet and replaces CSPDarknet53 in the Backbone network of the original YOLOv5s model with GhostNet. The CBL module in the Neck network is replaced by Ghost convolution, and the model can be compressed and simplified while the accuracy of the model is guaranteed. This provides a train of thought for the chili picking robot to detect and locate chilies in a complex field environment in real time and accurately.

(2) We proposed a lightweight GNPD-YOLOv5s model, which carried out channel pruning on the GN-YOLOv5s model to further reduce the complexity of the model, and used feature distillation technology to carry out distillation on the pruned model to compensate for the loss of detection accuracy of the pruned model.

(3) The depth information is introduced to establish a locating model to realize the three-dimensional spatial locating of chili fruit.

## 2. Related Work

When deploying the target recognition network in the picking robot, not only the computational complexity and the number of parameters of the model, but also the recognition accuracy of the model should be considered. Among the more common methods are network pruning [16], network parameter quantification [17], and knowledge distillation. They are based on the designed network model. Network pruning is a model optimization technology that removes redundant channels in weights. The compressed neural network can maintain faster running speed and lower computing cost. Network parameter quantization aims to reduce the space required for network weight storage by sacrificing the accuracy of parameters. Reasonable parameter quantization can reduce the volume of the network model exponentially on the premise of ensuring the accuracy of the network. Knowledge distillation [18] seeks to guide and simplify small models with low complexity through knowledge transfer in a relatively short time through a complex and huge deep model, and finally get a small-scale model with a simpler structure, reduced complexity, and improved reasoning performance.

In addition to compressing and quantifying some existing networks, some mature large-scale target recognition networks are also cut, and their network structures are adjusted to ensure that the network recognition accuracy is only reduced within a certain range; in this way, the network size can be greatly reduced. For example, Redomon designed Tiny-YOLOv2 [19] and Tiny-YOLOv3 [20], which are optimized and modified on the basis of the mature YOLOv2 and YOLOv3 recognition networks, At the expense of object recognition performance, the model size is greatly reduced. Combined with pruning and parameter quantification, these networks are also widely used in industry. However, in order to be more suitable for mobile scenarios, more and more lightweight networks are directly customized for mobile scenarios, such as Mobilenetv1-v3 [21,22,23] and Efficientdet [24] proposed by Google, GhostNet [25] proposed by Huawei, Shufflenet [26,27] and Squeezenet [28] proposed by Kuangshi Technology. These networks have built some ingenious neural network structures, leading to some new network design ideas. This approach can reduce the number of model parameters and improve the accuracy of network recognition, which is of great significance for the deployment of mobile terminals.

Binocular stereo vision is an important form of machine vision [29]. It is based on the principle of parallax and uses imaging equipment to obtain two images of the object being measured from different positions. It is a method to obtain three-dimensional geometric information on the object by calculating the position deviation between corresponding points of the image. It is applied to 3D reconstruction [30] and other works. Zhao et al. [31] used the Otsu and SSD algorithm to process the captured apple image and calculate the coordinates of the apple, and then convert the coordinate system to the robot arm coordinate system through affine transformation. This method completes the classification of each apple sample, and takes about 5.200 s. The fruit and vegetable picking robot can calculate the three-dimensional coordinates of fruits and vegetables more accurately and quickly through binocular stereo vision technology. Experts and scholars at home and abroad have made many research achievements in this regard. Wan et al. [32] developed a fruit picking system and proposed the DeepFruits algorithm. They used the deep network FasterRcnn to design a multi-class fruit recognition framework, and conducted experiments on multi-class crop datasets. The average recognition accuracy of the system was 92%. Type fruit recognition ability is weak. Jia et al. [33] combined binocular stereo vision technology and laser array technology to detect fruit, and this method has a very good effect on the resolution of overlapping strawberries. Luo et al. [34] used binocular stereo vision technology to detect grape picking points, and constructed a far-near stereo locating system to locate these points. The results showed that the judgment effect of this algorithm was good. According to the above literature, it is very effective to use binocular cameras to obtain the three-dimensional coordinates of fruits for locating fruit, and the approach is widely applicable. Therefore, it is feasible to achieve chili location by calculating the three-dimensional coordinates of chili fruits with binocular cameras. At present, the main problems are the method of recognizing the chili and the method of obtaining the two-dimensional coordinates of the chili fruit image. Only by determining the two-dimensional coordinates of chili fruits in the image can the corresponding three-dimensional coordinates be solved by binocular stereo vision technology.

## 3. Problem Statement

The main body of YOLOv5 model is basically composed of a Cross Stage Partial (CSP) structure. Since most of the convolution operations and parameters in the model are concentrated in this module, we first improve the CSP module. In order to ensure the accuracy of overlapping chili fruit recognition and reduce the computational loss of the YOLOv5 network, we propose a GN-YOLOv5s model for chili fruit detection based on feature distillation fused with GhostNet.

Firstly, in order to ensure the accuracy of overlapping chili fruit recognition and reduce the computational loss of the YOLOv5 network, we propose a GN-YOLOv5s model for chili fruit detection based on feature distillation fused with GhostNet. The GN-YOLOv5s network is divided into two parts: Backbone and Head, as shown in Figure 1. The backbone network of the model is a combination of GhostNet structure, convolution structure and SPP structure; The head network is composed of a feature pyramid (FPN) structure and a PAN structure. The SPP structure in the backbone divides the feature map into different spatial regions on different scales, then calculates the feature vector on each region, and finally combines all the calculated features. The advantage of the SPP structure is that it increases the receptive field of feature extraction, obtains the most important context features, and does not lead to speed reduction. The head network of the GN-YOLOv5s model consists of an FPN structure and a PAN structure.

The FPN structure conveys strong semantic features from the top down, while the PAN structure conveys strong location information from the bottom up, which greatly enriches the representation content of features and improves the performance of recognition.

Secondly, In order to reduce network complexity, improve recognition efficiency, and facilitate deployment to embedded devices with low computational power, the GNPD-YOLOv5s model is proposed as shown in Figure 2. GNPD-YOLOv5s consists of four stages: data enhancement, sparse training, pruning, and distillation. GhostNet is used as the backbone network of the YOLOv5s model, loading the enhanced data of mixed scenes for sparse training, clipping the model channel according to the pruning strategy in Section 3.2.1, and taking the GNPD-YOLOv5s model as the student model, distilling it with the following loss function, and then fine-tuning the training to obtain the final model. The original YOLO loss function includes target, classification and coordinate frame losses, and the overall loss function is:(1)$$L_{YOLO}=f_{obj}(o_{i}^{gt},{\hat{o}}_{i}{\overset{}{\hat{o}}}_{i})+f_{class}(p_{i}{}^{gt},\overset{}{{\hat{p}}_{i}})+f_{bb}(b_{i}^{gt},\overset{}{{\hat{b}}_{i}}),$$
where ${\overset{}{\hat{o}}}_{i}$, $\overset{}{{\hat{p}}_{i}}$, $\overset{}{{\hat{b}}_{i}}$ respectively represent the target, category probability and coordinate box corresponding to the student model, $o_{i}^{gt}$, $p_{i}{}^{gt}$ and $b_{i}^{gt}$ are their real values, and $f_{obj}$, $f_{class}$ and $f_{bb}$ are the loss functions of the target, category and coordinate box, respectively.

Considering that YOLO is a single-stage detector, which includes the prediction of the background bounding box, the standard distillation method will transfer the prediction of the teacher model on the background box to the student model, affecting the latter’s training on the target box. Therefore, the distillation loss is converted into a target scale function. Only when the target value predicted by the teacher model is high, the class probability and coordinate frame can be learned during distillation. The target loss function is modified as shown in Formula (2):(2)$$f_{obj}^{Comb}(o_{i}^{gt},{\overset{}{\hat{o}}}_{i},o_{i}^{T})=f_{\mathrm{obj}}(o_{i}^{gt},{\overset{}{\hat{o}}}_{i})+\lambda_{D}\cdot f_{obj}(o_{i}^{T},{\overset{}{\hat{o}}}_{i}),$$
where the first item $f_{\mathrm{obj}}(o_{i}^{gt},{\overset{}{\hat{o}}}_{i})$ is the target loss, the second item $\lambda_{D}\cdot f_{obj}(o_{i}^{T},{\overset{}{\hat{o}}}_{i})$ is the distillation loss, $\lambda_{D}$ is the weight coefficient, and $o_{i}^{T}$ is the prediction target of the teacher model. The classification loss function of the of the student model is shown in Formula (3), and similarly, the coordinate box loss function is shown in Formula (4). The total loss function of final distillation is shown in Formula (5), which includes the loss of coordinate frame, classification and target. L2 loss is used as the basic distillation function.
(3)$$f_{class}^{Comb}(p_{i}^{gt},{\overset{}{\hat{p}}}_{i},p_{i}^{T},{\hat{o}}_{i}^{T})=f_{class}(p_{i}^{gt},{\overset{}{\hat{p}}}_{i})+{\hat{o}}_{i}^{T}\cdot \lambda_{D}\cdot f_{class}(p_{i}^{T},{\overset{}{\hat{p}}}_{i}),$$
(4)$$f_{bb}^{Comb}(b_{i}^{gt},{\overset{}{\hat{b}}}_{i},b_{i}^{T},{\hat{o}}_{i}^{T})=f_{bb}(b_{i}^{gt},{\overset{}{\hat{b}}}_{i})+{\hat{o}}_{i}^{T}={\overset{}{{}^{}\hat{o}}}_{i}^{}\cdot \lambda_{D}\cdot f_{bb}(b_{i}^{T},{\overset{}{\hat{b}}}_{i}),$$
(5)$$L_{Distillation}=f_{bb}^{Comb}(b_{i}^{gt},{\overset{}{\hat{b}}}_{i},b_{i}^{T},{\hat{o}}_{i}^{T})+f_{class}^{Comb}(p_{i}^{gt},{\overset{}{\hat{p}}}_{i},p_{i}^{T},{\hat{o}}_{i}^{T})+f_{obj}^{Comb}(o_{i}^{gt},{\overset{}{\hat{o}}}_{i},o_{i}^{T}),$$

In Formula (3), the first item is the original classification loss, the second item is the distillation classification loss, $\lambda_{D}$ is the weight coefficient, and ${\hat{o}}_{i}^{T}$ is the prediction goal updated by teacher model.

### 3.1. GhostNet Module

In the task of chili fruit recognition, in order to refine the model and release the computing resources, the lightweight GhostNet module is introduced into the YOLOv5s model to reduce the number of parameters and computation of the model, and accelerate the reasoning speed of the original network, with the aim of making the model lightweight.

The GhostNet module is the part of Ghost convolution. Firstly, a small number of feature maps are generated by conventional convolution with less computation. Then, a new similar feature map is generated by further using a small number of feature maps through linear operation. Finally, the information in the two groups of feature maps is combined as all feature information, as shown in the Figure 3. Ghost volume integration includes three steps: regular convolution, Ghost generation and feature map splicing.

(1) Supposing that the size of the input feature map is H×W×C, the size of the output feature map is $H^{\prime }\times W^{\prime }\times n$, and the size of the convolution kernel is k×k. The feature map $Y_{H^{\prime }\times W^{\prime }\times m}$ is obtained by conventional convolution, and the calculation amount of this part is about equal to $H\times W\times c\times m\times W^{\prime }\times H^{\prime }$.

(2) The $\phi_{i}$ operation is used to generate the Ghost feature map from the feature map of each channel in feature map $Y_{H^{\prime }\times W^{\prime }\times m}$. In the linear transformation ($\phi_{i}$), if the number of channels of the characteristic graph is m, the number of transformations is s, and the number of new characteristic graphs finally obtained is n, then the equation can be obtained as shown in Formula (6).
(6)n=m·s.

Since there is an identity transformation in the transformation process of the Ghost module, the actual effective transformation quantity is (s−1). According to Formula (6), we can get:(7)$$m\times \left(s-1\right)=\frac{n}{s}\left(s-1\right).$$

(3) Finally, the feature map obtained in the first step is spliced with the Ghost feature map obtained in the second step (Identity connection) to get the final result.

Taking the above into account, when the number of channels of the output feature map is much greater than the number of channels of the intrinsic feature map (*n* >> *m*), we can obtain the calculation amount ratio of the ordinary convolution and Ghost module.
(8)$$r_{s}=s$$

Compared with the conventional convolution directly, the calculation amount of the Ghost convolution is greatly reduced: a simple linear transformation can produce most of the characteristic information. Using these features of Ghost convolution, this paper designs the Ghost module as the convolution layer in the backbone network, so that the overall network structure has the ability of multi-scale detection while maintaining the depth, so that the model is more suitable for the detection of chili fruit with the overlapping occlusion phenomenon.

### 3.2. Model Compression

In order to further compress the model parameters and improve the recognition accuracy of the YOLOv5s model fused with GhostNet, a model compression scheme of first pruning and then distillation is proposed to optimize the YOLOv5s model fused with GhostNet.

#### 3.2.1. Channel Pruning

Pruning can reduce the calculation amount and storage scale of the model, and speed up model inference. The designed pruning flow chart is shown in Figure 4. Firstly, we sparsely train the YOLOv5s model integrating GhostNet, then prune the non key channels of the model, and fine-tune the pruned network. Since too high a pruning rate will lead to the degradation of model accuracy, we set a low pruning rate to conduct pruning, then fine-tune the training and repeat the process many times to achieve the desired model compression result.

In this paper, the scaling factor γ of BN layer is used as the index to evaluate the channel importance. The size of γ is positively related to the channel importance. The calculation formula of the BN layer is as follows:(9)$$Z_{out}=\gamma .\overset{\wedge }{Z}+\beta ,$$
(10)$$\overset{\wedge }{Z}=\frac{Z_{m}-\mu }{\sqrt{\delta^{2}+\varepsilon }},$$
where γ and β are the normalized parameters of BN layer, $Z_{m}$ and $Z_{out}$ represent the inputs and outputs of BN layer respectively, μ and δ represent the mean and variance of BN layer respectively, and ε represents a small constant to prevent the denominator from being 0. Formula (9) shows that when γ approaches 0, the output is independent of the input, so the relationship between the inputs and outputs of these channels can be removed to reduce the amount of model parameters and calculations.

(1)Sparse training

In sparse training, an L1 regular penalty term is introduced to the parameter γ of BN layer. The sparse training loss function is shown in Formula (11).
(11)$$L=\underset{\left(x,y\right)}{\sum }l\left(f\left(x,W\right),Y\right)+\lambda \underset{\gamma }{\sum }g(\gamma ),$$
where the first term $\underset{\left(x,y\right)}{\sum }l\left(f\left(x,W\right),Y\right)$ is the loss function defined by YOLOv5s model integrating GhostNet. W represents the weight of the model, and (x, y) represents the input matrix and label. The second term $\lambda \underset{\gamma }{\sum }g(\gamma )$ is used to constrain the regular term penalty of γ, where λ is used to balance the two losses, which is called the sparse rate. Function g(*) is the L1 regularization, adding the L1 regularized loss function to drive γ towards the value of 0 to achieve the purpose of sparseness for γ.

(2)Channel pruning

Channel pruning is undertaken by cutting the channel whose γ is smaller; that is, the input and output of the pruned channel are removed so that a compression model with less parameter calculation is obtained. It is shown in the left part of Figure 5 that both Ci2 and Ci4 among the scaling factors approach zero. Therefore, the input and output of the corresponding channels of Ci2 and Ci4 are trimmed out in the model after pruning.

#### 3.2.2. Knowledge Distillation

After pruning of the YOLOv5 model, the complexity of the model has been greatly reduced, but this is also accompanied by a certain loss of accuracy. Knowledge distillation can improve the recognition accuracy of the model. The core idea of knowledge distillation is to let the teacher network guide the student network training, and the student network can improve the accuracy through the teacher network’s reference. This paper adopts knowledge distillation at the output layer. We take the original YOLOv5s model chili recognition model as the teacher network, and the pruned YOLOv5s model integrated with GhostNet as the student network. Through the complete output of the original YOLOv5s model, the pruned YOLOv5s model is integrated with the GhostNet master reasoning mode of the original YOLOv5s, so as to achieve the purpose of improving the model accuracy. The chili recognition method based on GNPD-YOLOv5s is designed by combining the above lightweight approach and the working method of the detector.

#### 3.2.3. Location of Chili Fruit

Since the chili image taken by the camera belongs to two-dimensional space, and its feature analysis is carried out in a two-dimensional plane, it is necessary to establish the pose features of the chili fruit to be picked through the conversion from two-dimensional space to three-dimensional space. In this paper, while the YOLOv5 model is improved to identify chili, the central coordinates of the recognition box are calculated and stored locally. The calculation process of the central coordinates of the recognition box is as follows:(12)$$center=(\frac{X_{\min}+X_{\max}}{2},\frac{Y_{\min}+Y_{\max}}{2}),$$
where $X_{\min}$, $X_{\max}$, $Y_{\min}$ and $Y_{\max}$ represent the image coordinates of the upper left corner and lower right corner of the recognition box, respectively. Since the position of the object in space is represented by the world coordinate system, its three-dimensional information is mapped to the image by the camera model. In our work, the image coordinate system, pixel coordinate system and camera coordinate system are established. The pixel coordinate system is established in the upper left corner of the image to describe the position of the pixel in the image. The image coordinate system takes the center point of the image as the origin; the coordinate axis is parallel to the pixel coordinate system, and the position of the pixel in the image is expressed in physical length units. The relationship between the image coordinate system and the pixel coordinate system is shown in Figure 6.

In order to realize the conversion from two-dimensional space to three-dimensional space, this paper uses the binocular camera vision system. The left and right cameras of the binocular camera shoot the same object from different angles. Then, through the triangulation theorem, as shown in Figure 7, the three-dimensional space position of the object is obtained by calculating the parallax. The projection of any point P (X, Y, Z) on the left and right camera planes in the camera coordinate system is $p_{l}\left(x_{l},\;y_{l}\right)$ and $p_{r}\left(x_{r},\;y_{r}\right)$ respectively. If $y_{l}=y_{r}$, according to the triangulation principle, we obtain Formula (13):(13)$$\frac{Z}{f}=\frac{X}{f}=\frac{X-B}{y_{r}}=\frac{Y}{y_{l}}=\frac{Y}{y_{r}},$$
where f is the focal length of the camera. The position of an object in three-dimensional space is calculated by parallax.

## 4. Experiment

### 4.1. The Chili Datset

#### Acquisition of Image Data

We took Erjingtiao chili under open field cultivation as the research object, and all the images needed for the experiment were collected from the chili planting base of Guizhou Academy of Agricultural Sciences. In open cultivation mode, row spacing of chili trees was about 10 cm and plant spacing was about 40 cm. In order to accurately reflect the complexity of chili growth and environment, chili images on sunny and cloudy days were collected. Filming sessions included morning, noon and afternoon. The images under different lighting and angle conditions were collected, including: no shade, no overlap chili image, fruit partially overlapping chili image, leaf shade chili image, branch shade chili image and mixed leaf shade chili image. The acquisition device is the built-in camera of the mobile phone, and the contrast, saturation and sharpness are set to standard mode. A total of 2000 images were collected, including the following conditions: chilies obscured by leaves and branches, mixed chilies obscured by overlapping chilies, natural light angle, backlight angle and side light angle. In addition, cloud, shadow, highlight, low light, reflection and other environmental factors are considered in image acquisition. There were 785 overhead images (462 in positive light, 323 in backlight) and 1215 horizontal images (871 in positive light, 344 in backlight). Figure 8 shows part of the images we collected. Labeling software was used to manually annotate the chili in the image, and the corresponding XML annotation file was generated. We draw the external rectangular box of the chili target to realize manual labeling of the chili. The image is labeled according to the smallest rectangle around the chili to ensure that the rectangle contains as little background area as possible. There are 1600 training sets, 200 verification sets and 200 test sets in a ratio of 8:1:1, The data set partition is shown in Table 1. In order to enrich the training set of image data, data enhancement is carried out on the data set to better extract the features of different label categories and avoid overfitting of the training model.

### 4.2. Analysis of Experimental Results

The experimental training process was carried out under the environment of Ubuntu 18.0 and CUDA 11.0. GPU configuration: NVIDIAGeForceRTX3090, 24 GB video memory, and GPU was called for training. In all experimental training parameter settings, the input image size is 640 × 640. The optimizer adopts the SGD optimizer of driving quantity. The initial learning rate is set to 0.001, the batch size is 16, and a total of 100 rounds of training are conducted. In order to objectively evaluate and compare the performance of the improved lightweight model, the model performance is evaluated from the aspects of model complexity and accuracy. The evaluation indexes include model parameters, floating point operands, model size, average accuracy(mAP), and model reasoning speed FPS.

#### 4.2.1. Sparse Training

Figure 9 shows the distribution of BN layer scaling factors when the sparsity is set to 0.01, 0.007, 0.005, and 0.003, respectively. From the figure, it can be seen that when the sparsity is smaller, the values of scaling factors distributed around 0 are less, and the effect of sparse training is not achieved. When the sparsity is larger, more channels with lower importance in the model can be clipped, but the accuracy of the model will be reduced at this time. The sparsity of the combined model and the detection accuracy were finally set to a sparsity of 0.005.

#### 4.2.2. Comparative Experimental Analysis of Recognition Models

In order to verify the performance of the YOLOv5s chili recognition model based on the lightweight fusion GhostNet of model compression scheme, this model is compared with the chili recognition model trained in the YOLOv5s model, RetinaNet, Faster R-CNN, SSD model and other mainstream algorithms.

Table 2 compares the recall rate, accuracy and detection speed of various algorithms in the chili data set created by us. It can be seen from Table 2 that the detection effect and reasoning speed are better than those of the RetinaNet, Faster R-CNN and SSD algorithms.

To verify the lightness of the GNPD-YOLOv5s model, we compare the computational and parametric quantities of several mainstream target detection models with the GNPD-YOLOv5s model, and the results are shown in Figure 10 and Figure 11.

Figure 10 and Figure 11 show the comparison of the parameters and computational effort of the GNPD-YOLOv5s chili detection model, other mainstream target detection models, and the original YOLOv5s model. It can be seen from the figures that the number of parameters and the computational effort are significantly reduced after model compression.

After the original YOLOv5s model is lightened, the feature information is lost. Therefore, we visualized the features extracted from the GNPD-YOLOv5s model and the original YOLOv5s model, as shown in Figure 12. In general, the feature extraction of the two models is similar. The chili lines extracted by the GNPD-YOLOv5s model were obvious, and the chili outline was retained in the feature map.

Figure 13 is the average accuracy(mAP) of the GNPD-YOLOv5s model. The model training converges around 80. Figure 14 shows the comparison between the GNPD-YOLOv5s model recognition model and the original YOLOv5s model. As can be seen from the figure, after compression, the model can still maintain a higher recognition accuracy under the condition where the chili is slightly shielded. In dense occlusion environments, accuracy is slightly reduced, as shown by the green marks in Figure 14f. However, the compressed model has high real-time performance and reasoning speed, and the requirements for equipment are reduced.

#### 4.2.3. Ablation Experiment

We tested the effect of introducing the lightweight Ghostnet and the model compression, respectively. From Table 3, we can see that the YOLOv5s lightweight model with Ghostnet as the backbone feature network has a much lower complexity after pruning. Figure 11 shows the number of parameters of the original YOLOv5s model and GNPD-YOLOv5s, and the number of parameters of the GNPD-YOLOv5s model are about 60.9% of those of the YOLOv5s model. It can be seen from Table 3 that the YOLOv5s lightweight model with Ghostnet as the backbone feature network has greatly reduced its complexity after pruning, and the model size is about 53.1% of YOLOv5s; the number of floating-point operations is about 59.1% of YOLOv5s, indicating that the introduction of Ghostnet can effectively reduce the complexity of the model. The effect of channel pruning on reducing the complexity of the model is verified. The accuracy of the model increased by 1.3% after knowledge distillation, which verified the effectiveness of knowledge distillation in improving the accuracy of the model.

### 4.3. Three Dimensional Space Locating of the Chili Target

#### 4.3.1. Spatial Localization of Chili

We use the IntelRealsenseD455i (Figure 15) depth camera as the spatial locating device for chili fruit. This depth camera can simultaneously obtain color images and depth images, and obtain depth information through infrared sensors. The color image resolution is 1280 × 720.

The location function is embedded into the recognition model to establish the real-time recognition and location model of chili. First, we call the depth camera to obtain the depth image and color image. Because the field of vision of the depth image and color image is different, it needs to be displayed as a color image finally. Therefore, the depth image is first aligned to the color image to ensure the same field of vision. When the recognition model detects the chili fruit, it returns the two-dimensional coordinates of the image center point, calculates the three-dimensional coordinates of the main stem growth point through the relevant parameter formula and other information, and displays the coordinate information in the upper left corner of the window. The overall process is shown in Figure 16.

In the locating process, the crop fruit will move 20 mm away from the camera each time, and it will be positioned once, mainly referring to the spatial depth of the chili fruit. The initial position of the chili fruit is (100 mm, 100 mm, 100 mm). The results of 8 times of locating are shown in Table 4.

The locating method is tested in the practical environment, and multiple groups of X, Y, Z coordinates are obtained, The error is calculated by comparing with the actual value. The absolute error can be obtained by subtracting the detected value from the actual value.

In the eight locating processes, the locating error does not exceed 1.84, and fluctuates several times around 0, which proves that the locating system has both accuracy and stability.

#### 4.3.2. Chili Ranging with Different Degrees of Shading

We placed the chili at a distance of 0.6 m from the camera, divided the shielding area of the chili from 20% to 80% according to the proportion of 10% intervals, and studied the influence of different shielding areas on the ranging accuracy of the chili. The ranging results are shown in Table 5. The statistical results in Table 5 show that when the occlusion area is 20–30%, the network can detect the complete chili target, and the measurement results are consistent with the real situation. The ranging results are about 0.608 m, and the error is about 0.08 m. When the occlusion area is 40% to 50%, the center of the prediction box cannot fall to the chili target, so the error is large, the measurement result is about 0.553 m, the error is about 0.047 m. When the shielding area was 50% to 70%, the center of the prediction frame fell to the surface of the chili, so the positioning accuracy was improved, and the measurement result was about 0.562 m. The error is about 0.038. When the shielding area is greater than 80%, the chili cannot be identified.

## 5. Conclusions

We propose a lightweight GNPD-YOLOv5s model for chili recognition and location. This model first uses Ghostnet lightweight network as the backbone network to realize the preliminary lightweight chili recognition model. Secondly, channel pruning is applied to the model, which greatly reduces the number of model parameters. Finally, the accuracy of the lightweight model was improved through knowledge distillation, and the final GNPD-YOLOv5s model was obtained. We built our own chili data set and verified the model on the chili data set. The experimental results show that the recognition accuracy and recognition speed of GNPD-YOLOv5s model reach 92.9% and 115.74 frames per second, respectively, meeting the requirements of a chili picking robot for chili recognition. The binocular camera is used to conduct three-dimensional locating in the actual chili plant scene. Through many experiments, the maximum locating error is not more than 1.84 mm, which meets the action planning requirements of the manipulator of the picking robot.

## Figures and Tables

**Figure 1 sensors-23-03408-f001:**
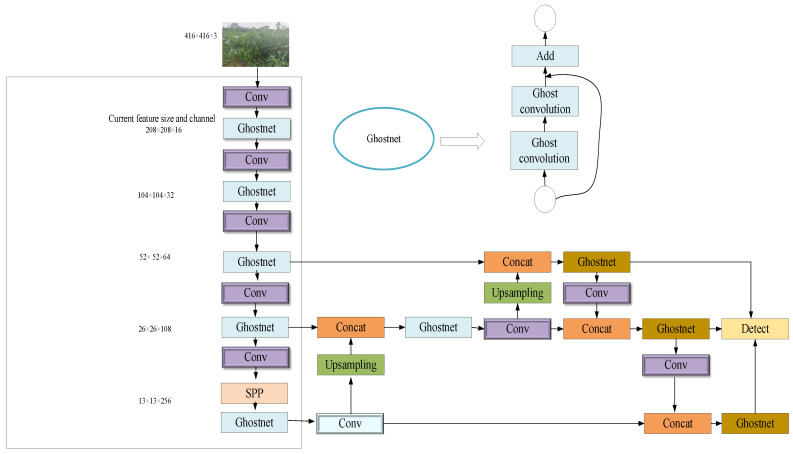
GN-YOLOv5s network architecture.

**Figure 2 sensors-23-03408-f002:**
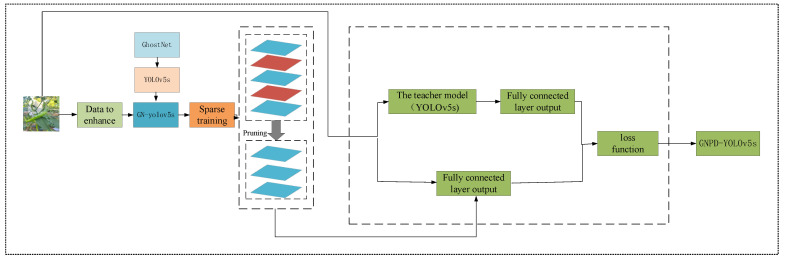
GNPD-YOLOv5 model.

**Figure 3 sensors-23-03408-f003:**
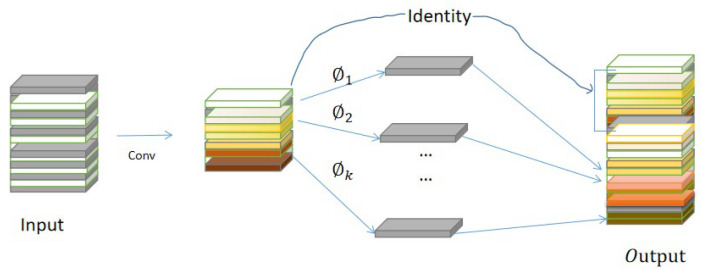
Schematic Diagram of Ghost Convolution Process.

**Figure 4 sensors-23-03408-f004:**
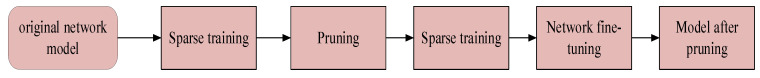
Flow chart of model pruning.

**Figure 5 sensors-23-03408-f005:**
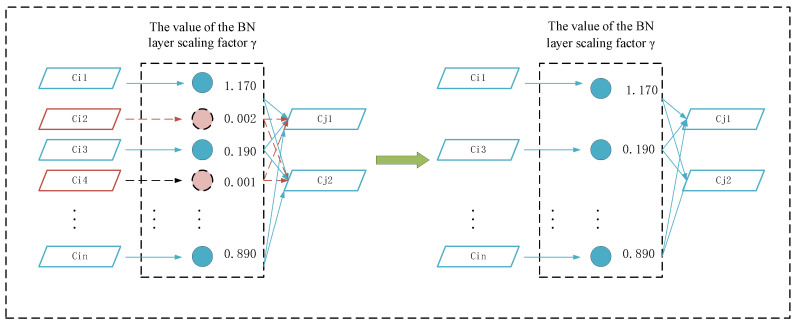
Schematic diagram of channel pruning.

**Figure 6 sensors-23-03408-f006:**
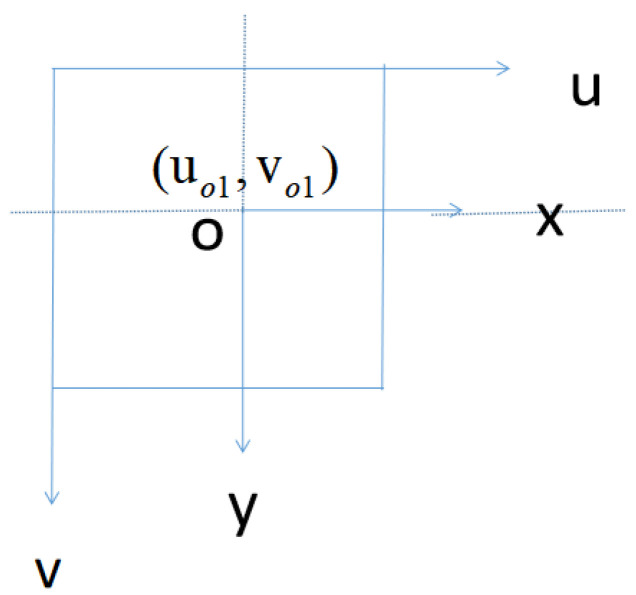
Relationship between image coordinate system and pixel coordinate system.

**Figure 7 sensors-23-03408-f007:**
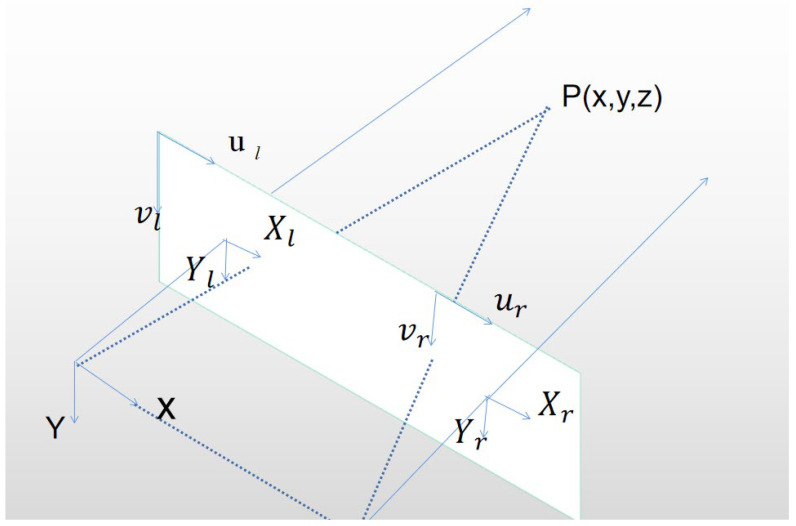
Binocular stereo locating model.

**Figure 8 sensors-23-03408-f008:**
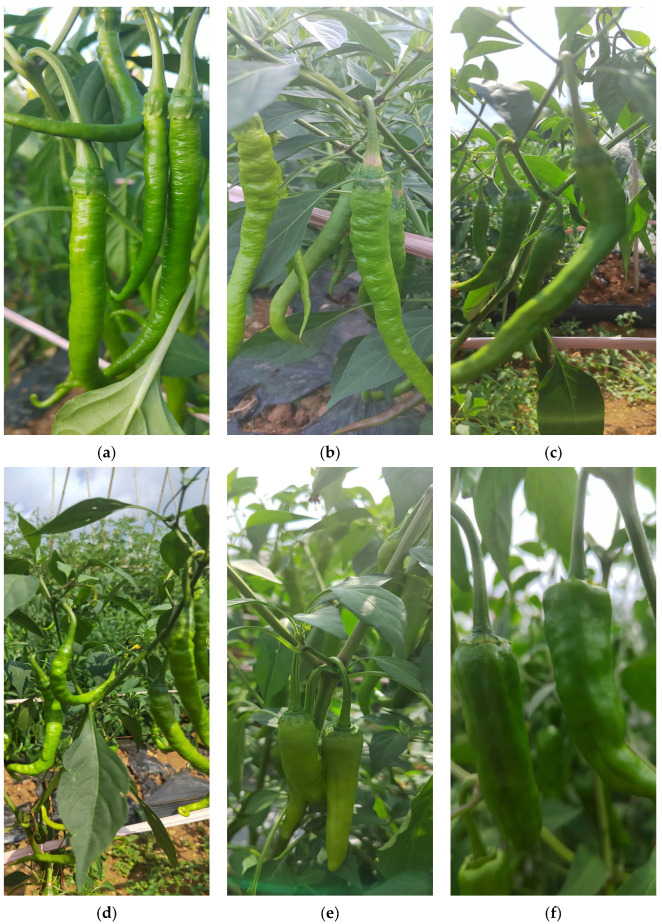
Pictures of chilies in different environments. (**a**,**b**) are pictures of chili under different degrees of occlusion. (**c**,**d**) are chili images under positive light. (**e**) chili image under backlight conditions. (**f**) chili image under cloudy conditions.

**Figure 9 sensors-23-03408-f009:**
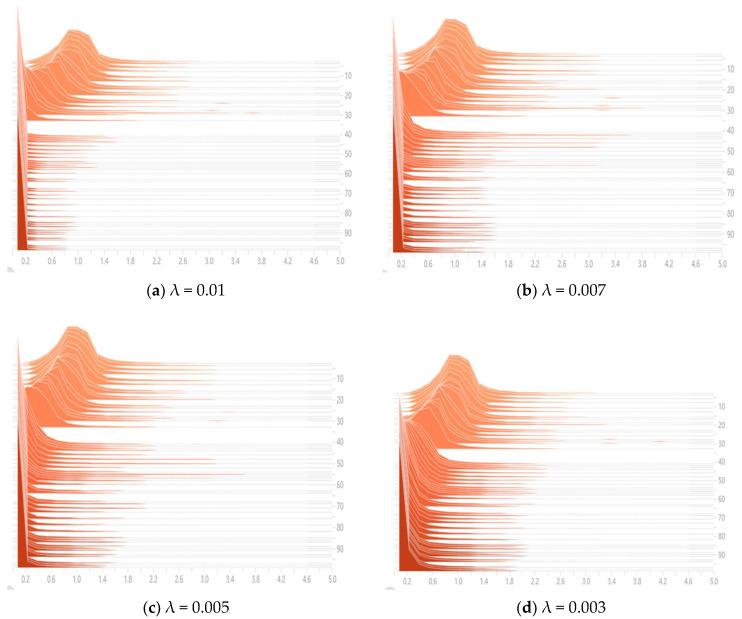
Distribution of sparse training scaling factor. (**a**) is the distribution of the BN layer scaling factor when the sparse rate is 0.01; (**b**) is the distribution of the BN layer scaling factor when the sparse rate is 0.007; (**c**) is the distribution of the BN layer scaling factor when the sparse rate is 0.005; (**d**) is the distribution of the BN layer scaling factor when the sparse rate is 0.003.

**Figure 10 sensors-23-03408-f010:**
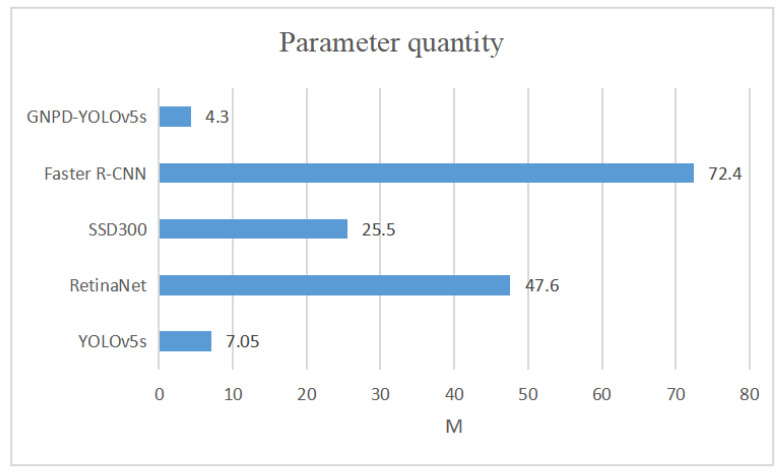
Comparison of parameters between GNPD-YOLOv5s chili detection model and other mainstream models.

**Figure 11 sensors-23-03408-f011:**
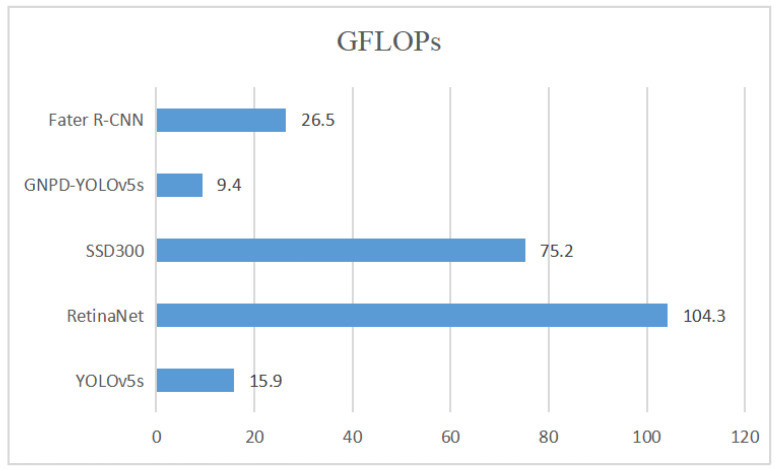
Comparison of GFLOPS amount between GNPD-YOLOv5s chili detection model and other mainstream models.

**Figure 12 sensors-23-03408-f012:**
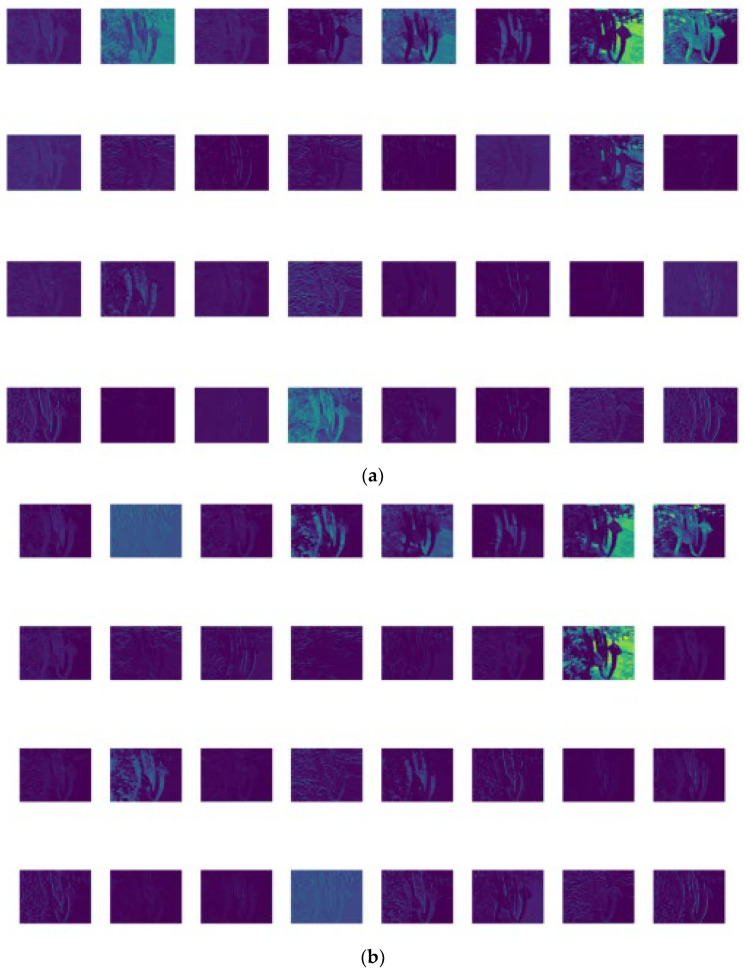
(**a**) Feature map extracted by YOLOv5s; (**b**) GNPD-YOLOv5s used to extract feature map.

**Figure 13 sensors-23-03408-f013:**
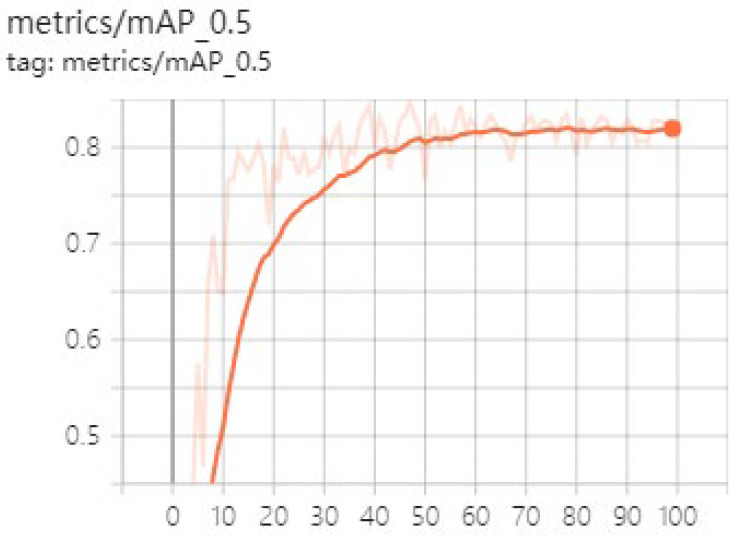
Average precision curve.

**Figure 14 sensors-23-03408-f014:**
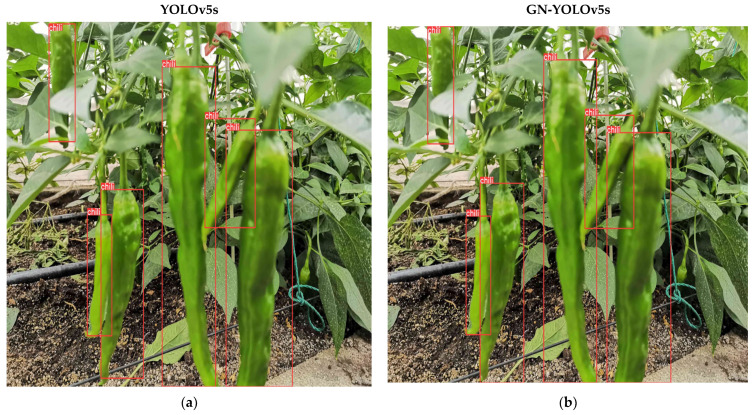
Comparison between original YOLOV5s and GNPD-YOLOv5s model. (**a**) chili detection of the original YOLOv5s model; (**b**) GN-YOLOv5s was used for comparison of chili detection; (**c**) Detection results of the original YOLOv5s model in light occlusion environment; (**d**) Comparison of GNPD-YOLOv5s models in light occlusion environment; (**e**) Detection results of the original YOLOv5s model under severe occlusion; (**f**) Comparison of GNPD-YOLOv5s model in severely occluded environment. (Green marks represent missed chilies).

**Figure 15 sensors-23-03408-f015:**
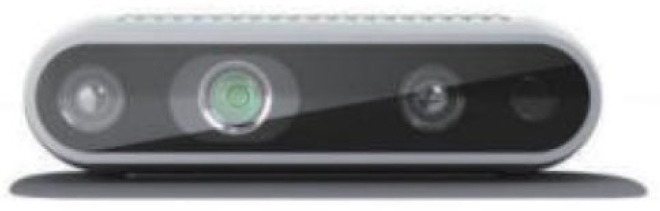
Intel realsense D455i.

**Figure 16 sensors-23-03408-f016:**
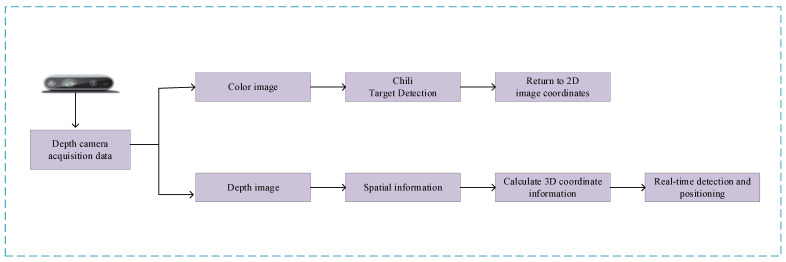
Real time recognition model.

**Table 1 sensors-23-03408-t001:** Occlusion and quantity of datasets.

Dataset	Number of Unoccluded Samples	Number of Occluded Samples	The Total Number
Training set	5324	2671	6995
Validation set	673	291	964
Test set	644	235	879

**Table 2 sensors-23-03408-t002:** Performance comparison of mainstream recognition algorithm models.

Model	Recall Rate	MAP	Inference Speed/(ms/Frame)
RetinaNet	0.863	0.745	37
Faster R-CNN	0.942	0.831	162
SSD300	0.792	0.769	32
SSD512	0.812	0.774	65
GNPD-YOLOv5s	0.927	0.869	14

**Table 3 sensors-23-03408-t003:** Ablation experiment.

Model	Reall Rate	Floating Point Operand/ GFLOPs	Model Size/MB	MAP	Inference Speed/ (ms/Frame)
YOLOV5s	0.942	15.9	14.74	0.892	29
GN-YOLOv5s	0.921	13.2	12.86	0.873	18
GN-YOLOv5s + Channel pruning	0.915	9.4	7.84	0.856	16
GNPD-YOLOv5s	0.927	9.4	7.84	0.869	14

**Table 4 sensors-23-03408-t004:** 3D locating of chili fruits.

Experiment Times	Theoretical Position	Locating Result of Moving 20 mm	Error/mm
1	(100 mm, 100 mm, 100 mm)	(100.62 mm, 100.54 mm, 100.06 mm)	0.06
2	(100 mm, 100 mm, 120 mm)	(100.58 mm, 100.64 mm, 120.12 mm)	0.12
3	(100 mm, 100 mm, 140 mm)	(100.55 mm, 100.29 mm, 140.08 mm)	0.08
4	(100 mm, 100 mm, 160 mm)	(100.69 mm, 100.78 mm, 161.62 mm)	1.62
5	(100 mm, 100 mm, 180 mm)	(100.81 mm, 100.96 mm, 180.24 mm)	0.24
6	(100 mm, 100 mm, 200 mm)	(100.027 mm, 100.45 mm, 200.43 mm)	0.43
7	(100 mm, 100 mm, 220 mm)	(100.95 mm, 100.43 mm, 220.16 mm)	0.16
8	(100 mm, 100 mm, 240 mm)	(100.34 mm, 101.15 mm, 241.84 mm)	1.84

**Table 5 sensors-23-03408-t005:** Different shade area range results.

Experiment Times	Shielding Area (%)	Ranging Result (m)
1	20	0.608
2	30	0.609
3	40	0.553
4	50	0.562
5	60	0.574
6	70	0.586
7	80	0.547
8	90	Cannot be identified

## Data Availability

The data presented in this study are available on request from the corresponding author.

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
