# Peer review of "Real-Time Recognition and Localization Based on Improved YOLOv5s for Robot’s Picking Clustered Fruits of Chilies"

_sensors, 2023, doi:10.3390/s23073408_

Round 1

Reviewer 1 Report

This paper makes many improvements to the existing model to reduce the parameters of the model and improve the recognition efficiency, and conducts a comparative analysis with a variety of recognition models to characterize the superiority of the model. The author should make corrections in two aspects:

1) The expression of the contribution part is not clear enough, for example, the expression in the first point of the contribution is confusing.

2) Although the paper gives the comparison results of various models, it does not perform visual analysis of the high-order features extracted by the models. Please give a visualization of the high-level features extracted by these models to better reflect the superiority of your model.

Author Response

Dear Reviewer,

Thank you very much for your comments. These comments are valuable and helpful in revising and improving our paper, and they serve as a valuable guide for our research. We carefully considered these comments and made revisions that we hope to get your approval.

Point 1: The expression of the contribution part is not clear enough, for example, the expression in the first point of the contribution is confusing.

Response 1: Thank you for your insightful comments. In order to clearly express our contribution, lines 100 to 112 of the manuscript re-describe the work we have done.

Point 2: Although the paper gives the comparison results of various models, it does not perform visual analysis of the high-order features extracted by the models. Please give a visualization of the high-level features extracted by these models to better reflect the superiority of your model.

Response 2: Thank you for the detailed review. We did additional feature visualizations and presented them on lines 513 through 516 of the manuscript.

Yours sincerely,

Song Zhang

Reviewer 2 Report

In this work the authors proposed a real-time system for detecting peppers. Overall, the paper is well-written, however, I have following concerns the authors need to consider while submitting the revised manuscript.

1.      What is the significance of proposed work.

2.      The contribution of proposed framework is missing.

3.      The authors need to discuss what are research gaps and how proposed framework filled those gaps.

4.      Scale problem lies in the heart of object detector. How the proposed deals with the scale problem?

5.      The detail of dataset is missing.

6.      What is the purpose of Figure 10.

7.      Experiment section is very weak. The author needs to compare the proposed framework with other reference methods

8.      How many parameters are used.

9.       It is important to access the framework in terms of computational complexity.

Author Response

Dear Reviewer,

Thank you very much for your comments. These comments are valuable and helpful in revising and improving our paper, and they serve as a valuable guide for our research. We carefully considered these comments and made revisions that we hope to get your approval.

Point 1: What is the significance of proposed work.

Response 1: Thank you for your insightful comments. Our previous description was not clear enough, which has been redescribed and modified by red font in lines 90 to 98 of the manuscript.

Point 2: The contribution of proposed framework is missing.

Response 2: Thank you for the detailed review. In order to clearly express our contribution, We supplement it on line 100 to 112 of the manuscript.

Point 3: The authors need to discuss what are research gaps and how proposed framework filled those gaps.

Response 3: Thank you for your careful review. Aiming at the problem that existing target detection algorithms are relatively complex and have many parameters. We build a model and hope to achieve lightweight model while maintaining the detection accuracy. We supplemented our model and how to fill these gaps in lines 83-89 of the manuscript, and modified the manuscript in red font.

Point 4: Scale problem lies in the heart of object detector. How the proposed deals with the scale problem?

Response 4: Thank you for your valuable advice. In Section 3.1 of the manuscript, we discussed the calculation amount after replacing the original YOLOv5 backbone network with Ghostnet module, and supplemented the comparison of the number of model parameters in the experiment. In Section 3.2 of the manuscript, the model size problem of the model after channel pruning is described.

Point 5: The detail of dataset is missing.

Response 5: Thank you for your careful reading. We supplemented the dataset information and we described in lines 377 to 459 of the manuscript.

Point 6: What is the purpose of Figure 10.

Response 6: Thank you for the detailed review. The purpose of Fig.10 is to show that in sparse training, if the sparse rate is set too small, the proportion factor values which distributed near 0 are small, and high-intensity pruning cannot be carried out.  If the setting is too large, network performance will be affected, so we need to experiment to find a suitable value.  We supplement this question in lines 472 to 482 of the manuscript.

Point 7: Experiment section is very weak. The author needs to compare the proposed framework with other reference methods

Response 7: Thank you very much for your comments. We have supplemented experiments on the basis of previous studies. In lines 499-513 of the manuscript, we add a comparison experiment between the number of parameters and the amount of computation of our model and several mainstream target detection models. In lines 514 to 520 of the manuscript, we visualized feature extraction; In line 523 to 552 of the manuscript, we verify the feasibility of the model for several occlusion situations.

Point 8: How many parameters are used.

Response 8: We added an experiment on the number of contrast parameters to the manuscript, which was shown in Figure 10 of the manuscript.

Point 9: It is important to access the framework in terms of computational complexity.

Response 9: Thank you very much for your comments. We compared our model with several other mainstream target detection models in computational complexity, which is shown in Figure 11 of the manuscript.

Yours sincerely,

Song Zhang

Reviewer 3 Report

There are some alignment  problem page Figure 4 and 9  captions

Figure 9 vegetable is chilly but the article claim the work is for pepper .pepper is different  fruit /vegetable ,which should be checked

 figure size and quality to be improved for example figure number 11

  results for locating in overlapping occlusion state is not analysed. It must be analysed because the author claims the architecture is designed for solve this issue

Location accuracy can also be tested on overlapped images light medium over  ,high overlap and light overlap 

Author Response

Thank you very much for your comments. These comments are valuable and helpful in revising and improving our paper, and they serve as a valuable guide for our research. We carefully considered these comments and made revisions that we hope to get your approval.

Point 1: There are some alignment problem page Figure 4 and 9 captions.

Response 1: Thanks for your careful review, we have corrected the alignment problem between Figure 4 and Figure 9.

Point 2: Figure 9 vegetable is chilly but the article claim the work is for pepper .pepper is different fruit /vegetable ,which should be checked figure size and quality to be improved for example figure number 11.

Response 2: Our entire article was implemented in a chili field environment, and we have corrected this error throughout the article. In view of the quality problem in Figure 11, we have re-uploaded the picture with higher quality. Because of we added some experiments, Figure 11 is change to Figure 13.

Point 3: results for locating in overlapping occlusion state is not analysed. It must be analysed because the author claims the architecture is designed for solve this issue.

Response 3: Thank you for your insightful comments. We added an experiment on chile recognition in the overlapping state. We described the corresponding experiment and marked it with red in lines 523 to 551 of the manuscript.

Point 4: Location accuracy can also be tested on overlapped images light medium over high overlap and light overlap

Response 4: Thank you for your valuable comments. We supplemented experiments for chili localization with various degrees of overlap and we described them in lines 614 to 627 of the manuscript.

Yours sincerely,

Song Zhang

Round 2

Reviewer 2 Report

Thanks for considering and addressing my concerns

Reviewer 3 Report

All comments were addressed, but Point 2:  command was not addressed entirely still, the word pepper is available in figure(16) and the text (line no 89,502..)

Author Response

Dear Reviewer,

Thank you very much for your comments. These comments are valuable and helpful in revising and improving our paper, and they serve as a valuable guide for our research. We carefully considered these comments and made revisions that we hope to get your approval.

Point 1: All comments were addressed, but Point 2: command was not addressed entirely still, the word pepper is available in figure(16) and the text (line no 89,502.)

Response 1: Thanks for your careful review, In response to this error, we have followed up the correction throughout the manuscript. We have replaced "pepper" with "chili" in lines 82, 84, 85, 87, 89, 502 and 505, and marked it in green. In Figure 16 of the manuscript, we replace "pepper" with "chili".

Yours sincerely,

SongZhang
